# Long-Term Effects of 0.1 mg Recombinant-Human-Thyrotropin-Stimulated Fixed-Dose Radioiodine Therapy in Patients with Recurrent Multinodular Goiter after Surgery

**DOI:** 10.3390/diagnostics14090946

**Published:** 2024-04-30

**Authors:** Nicholas Angelopoulos, Ioannis Iakovou, Grigoris Effraimidis, Sarantis Livadas

**Affiliations:** 12nd Academic Nuclear Medicine Department, Academic General Hospital of Thessaloniki “AHEPA”, Aristotle University of Thessaloniki, 54636 Thessaloniki, Greece; iakovou@otenet.gr; 2Department of Endocrinology and Metabolic Diseases, Larissa University Hospital, Faculty of Medicine, School of Health Sciences, University of Thessaly, 41110 Larissa, Greece; grigoris.effraimidis@gmail.com; 3Department of Endocrinology and Metabolism, Rigshospitalet, Copenhagen University Hospital, 2100 Copenhagen, Denmark; 4Endocrine Unit, Athens Medical Centre, 65403 Athens, Greece; sarntis@gmail.com

**Keywords:** thyroid nodules, recombinant TSH, multinodular goiter, thyroidectomy

## Abstract

(1) Background: After thyroid malignancy is ruled out, treatment options for multinodular goiter patients include surgery, levothyroxine suppressive therapy, and 131-I therapy. Surgery effectively reduces goiter size but carries risks of surgical and anesthetic complications. 131-I therapy is the only nonsurgical alternative, but its effectiveness diminishes with goiter size and depends on iodine sufficiency. This study aimed to assess the efficacy and safety of 0.1 mg rhTSH as an adjuvant to a fixed dose of 131-I therapy in patients with a recurrence of large multinodular goiter, several years after the initial thyroidectomy. (2) Methods: 14 patients (13 females and 1 male), aged 59.14 ± 15.44 (range, 35–78 years) received 11mciu of 131-I, 24 h after the administration of 0.1 mg rhTSH. The primary endpoint was the change in thyroid volume (by ultrasound measurements) as well as in the diameter of the predominant nodule during a follow-up period of 10 years. Secondary endpoints were the alterations in thyroid function and potential adverse effects. (3) Results: A significant decrease in the volume of initial thyroid remnants (32.16 ± 16.66 mL) was observed from the first reevaluation (at 4 months, 23.12 ± 11.59 mL) as well as at the end of the follow-up period (10 years, 12.62 ± 8.76 mL), *p* < 0.01. A significant reduction in the dominant nodule was also observed (from 31.71 ± 10.46 mm in the beginning to 26.67 ± 11.05 mm). (4) Conclusions: Further investigation is needed since this approach could be attractive in terms of minimizing the potential risks of reoperation in these patients.

## 1. Introduction

Asymptomatic euthyroid patients with benign non-toxic goiter (NTG) and no compressive symptoms can be observed with clinical evaluations, laboratory evaluations, and thyroid imaging tests. The American Thyroid Association recommends a standard follow-up interval of 6–18 months for patients with NTG, which may be progressively lengthened if no significant changes are observed within the first 3–5 years [1]. Thyroid hormone treatment is not indicated in euthyroid individuals with nodular thyroid disease according to the recently published guidelines from the European Thyroid Association [2].

Total and near-total thyroidectomy is currently the preferred surgical treatment option for symptomatic nodular thyroid disease [3], while for diseases limited to one lobe, lobectomy/hemithyroidectomy is recommended [2]. In the 1980s and 1990s, however, the overwhelming majority of patients underwent subtotal or partial thyroidectomy, resulting in a high recurrence rate [4] and a new intervention in 2.5% to 42.5% of these patients.

After subtotal thyroidectomy, the treatment of choice is suppressive therapy (preventive administration of levothyroxine (LT4) to reduce serum TSH levels below normal). This approach has proved ineffective and has numerous drawbacks: only one [5] of four randomized trials [5,6,7,8] showed LT4 to be effective in this setting; it requires permanent treatment, and it may have undesirable effects on bone (demineralization) [9] and the heart (arrhythmias), particularly in older individuals [10].

Several partially thyroidectomized patients with recurrent large benign goiters may exhibit signs of pressure, such as dysphagia, neck rigidity, or a sense of airway obstruction. Such patients frequently require reoperation to alleviate their symptoms; however, the benefits and risks of a new surgical procedure must be carefully weighed, as the incidence of recurrence rises with age, affecting especially elderly patients who frequently suffer from comorbidities [11]. Significantly increased rates of permanent complications, such as hypoparathyroidism and recurrent laryngeal nerve dysfunction, are associated with new surgical interventions [11].

In these patients, radioiodine therapy (RAI) could be an alternative. NTG patients are hampered, however, by low isotope accumulation in inactive and partially suppressed areas surrounding the nodules. To rectify this issue, increasing the RAI dose or using recombinant human thyroid stimulating hormone (rhTSH) has proven effective [12,13]. Upon stimulation with rhTSH, the dormant thyroid tissue, which faintly concentrates radioiodine, reactivates and eventually amplifies the effect of the RAI in the gland, resulting in a further decrease in the volume of the goiter.

The combination of rhTSH and radioiodine is well tolerated [14], and the potential adverse effects are comparable to those observed in patients receiving radioiodine alone [15]. Patients may experience a transient increase in thyroid hormone levels within 48 h of radioiodine administration, resulting in moderate thyrotoxicosis. The acute adverse events reported include agonizing transient thyroiditis, thyroid enlargement, tracheal compression, and, frequently, heart-related symptoms [14,15].

Multiple studies have demonstrated that rhTSH doses as low as 0.01 enhance RAI, but the most effective dose with minimal side effects is 0.10 [16,17,18]. On the other hand, multiple studies have demonstrated that the administration of rhTSH before radioiodine therapy for volume reduction of nontoxic, nodular goiter may enable treatment with lower doses of radioiodine without reducing the radiation dose absorbed by the thyroid [16,17,18]. There is sufficient evidence in the literature [14,16,17,18,19] regarding the short-term efficacy and safety of radioiodine therapy with a low rhTSH dose (0.01 or 0.1 mg) in thyroid volume reduction. Still, there are little data regarding long-term follow-ups.

## 2. Aim

This study aimed to determine the durability of this beneficial effect in a solid cohort of patients with recurrent euthyroid nontoxic multinodular goiter after partial thyroidectomy, who were initially treated with a fixed small dose of RAI (11mCi) under 0.1 mg of rhTSH stimulation 10 years after therapy.

## 3. Materials and Methods

### 3.1. Study Participants

This was a retrospective, long-term follow-up study of our previous reports [20] on partially thyroidectomized patients with large recurrent multinodular goiter several years after thyroidectomy (mean, 8.32; range, 5–16 years). Patients initially eligible for enrollment were those admitted to our endocrine outpatient clinics with a recurrent large multinodular goiter resulting in cervical compression and/or cosmetic discomfort between January 2008 and December 2009. Due to the goiter’s size and local discomfort symptoms, a second surgery approach would have been the treatment of choice for such patients. However, when surgery was not feasible because of concomitant medical disorders, previous neck surgery, and/or personal aversions, study enrollment was offered. Subjects older than 18 years with a recurrent NTG and two or more nodules larger than 1 cm and the presence of goiter-related symptoms (i.e., pressure and/or cosmetic complaints) were eligible. Exclusion criteria included a history of cardiac failure or ventricular arrhythmias, previous malignant disease, or physical or psychiatric disorders suggestive of difficulties in adherence to the protocol. The initial setup included clinical examination, thyroid function tests [TSH, free-T4 (FT4), free-T3 (FT3)], 99 m Tc-pertechnetate scintigraphy, and ultrasonography. A fine-needle aspiration biopsy of the sonographically dominant and scintigraphically hypoactive nodules was performed to exclude malignancy. Thus, the study population comprised a selected group of 17 patients. All participants signed a consent form, and the study was approved by the Institutional Review Board (IRB protocol number: Ν2024/0121312) of the Hellenic Endocrine Network Institution, Athens, Greece.

### 3.2. Study Protocol

rhTSH in a fixed dose of 0.1 mg administered by intramuscular injection, and 24 h later, a fixed dose of 11 m Ci were administrated. A freeze-dried vial containing 0.9 mg of rhTSH (Thyrogen Genzyme, Cambridge, MA, USA) was reconstituted with 1 mL of sterile water and then diluted to a final concentration of 0.1 mg/mL with normal saline. Follow-up clinical examinations were scheduled frequently during the first 2 weeks (days 1, 3, 7, and 14 after RAI therapy) to evaluate potential adverse effects. Thyroid function tests (serum TSH, FT4, and FT3) were measured before RAI therapy at 4 and 12 months after RAI. Blood samples were collected in a fasting state at each examination and thyroid hormones were measured by means of an ultra-sensitive chemiluminescence immunoassay (Roche Diagnostics, Mannheim, Germany, normal ranges: 0.3–4.2 mIU/L for TSH, 0.93–1.71 ng/dL for FT4, and 2.53–4.42 pg/mL for FT3). Subsequent visits were scheduled yearly, as usually performed in all NTG patients, to investigate thyroid function. Finally, approximately 10 years after RAI treatment, clinical and laboratory data (ultrasound, TFTs, and LT4 dose) were collected for all participants. Iodine therapy was administered between August 2008 and July 2010.

### 3.3. Thyroid Size Estimation

Thyroid size was estimated at baseline and at 4 months, 12 months, and 10 years after 131-I therapy. In all patients, an ultrasonic scanning procedure with two-dimensional images was performed by one experienced operator (intraobserver variation coefficient, 4.7%) using a LOGIQ 9 ultrasound device mounted with a 12 MHz linear transducer (GE Medical Systems, Milwaukee, WI, USA). Since participants had previously undergone thyroid surgery, thyroid volume (TV) was calculated on the residual thyroid gland, and the sum of the volumes of the right lobe, left lobe, and isthmus was regarded as the total TV expressed in milliliters. The size of the dominant nodule in each patient was also calculated as the maximum diameter of three repeated measurements expressed in millimeters.

### 3.4. Statistical Analyses

Statistical analyses were performed using the SPSS statistical software program, version 16.0 (SPSS Inc., Chicago, IL, USA). Depending on the distribution of the data, parametric or nonparametric tests were used for analysis. Within-group change in absolute TV was assessed by Friedman’s test. Between-group differences in the relative change in TV were analyzed using an independent-sample *t*-test. To compare frequencies, the X2 test and Fisher’s exact test were used. The level of significance was 0.05. Values are given as mean ± standard deviation, SD, or median (25th and 75th percentiles).

## 4. Results

### 4.1. Thyroid Volume and Nodule Size

Of the 17 initially enrolled patients, 3 withdrew their informed consent before randomization, leaving 14 patients (1 male and 13 female patients) who completed final reassessment (the mean ± SD of the follow-up was 12.46 ± 1.16; range, 10.5–14 years). The TV significantly decreased at the end of the 10-year follow-up period (the mean ± SD of the TV at baseline vs. 10 yrs after RAI treatment: 32.16 ± 16.66 mL vs. 12.62 ± 8.76 mL, respectively, *p* < 0.001, Table 1).

The decrease was already significant at the first ultrasound evaluation at 4 months after RAI treatment (Figure 1, Table 2). A further improvement in TV was detected 12 months after RAI treatment; however, no statistical significance was further reached when compared with TV at 4 months after RAI treatment (*p* = 0.09, Table 2). There was a positive correlation between the decreased thyroid volume observed at 4 months and the volume of the thyroid remnant at the end of the study (r = 0.407, *p* = 0.014) (Figure 2).

Regarding the size of the dominant nodule, a significant reduction was observed at the end of the study (mean ± SD of the size of the dominant nodule at baseline vs. 10 yrs after RAI treatment: 31.71 ± 10.46 mm vs. 25.67 ± 11.05 mm, respectively, *p* < 0.05, Table 1). This reduction was already observed after 1 year of following up while no further decrease was detected thereafter (Figure 3). In one patient, there was no response to therapy, and an increase in the dominant nodule was detected during the follow-up (from 40 mm to 46 mm). In another patient, although the nodule decreased during the first year of following up from 24 to 16 mm, there was a substantial increase in the nodule size at the 10-year reassessment (32 mm). FNA was performed in both nodules to rule out malignancy (Bethesda II cytology). The comparisons of changes in the diameter of the dominant nodule are illustrated in Table 3.

The results of the multiple linear regression indicated that there was a moderate collective non-significant effect between the following: TV at 4 months (log values), TV at 12 months (log values), absolute difference of TSH values (post- and pre-injection), weight, BMI, age, and TV at final visit (log values) (F(1, 12) = 2.67, *p* = 0.128, R^2^ = 0.18, R^2^adj = 0.11).

### 4.2. Thyroid Function

Six out of fourteen patients (42.85%) were already hypothyroid at baseline. The baseline TSH for the eight euthyroid patients was 1.24 ± 0.59 mIU/L (mean ± SD, min 0.6 and max 2.49) and FT4 1.27 ± 0.27 ng/dL (mean ± SD). In the first year of following up, 2 more patients became hypothyroid (in total 8/14, 57.14%), while at the end of the study, permanent hypothyroidism requiring levothyroxine therapy had developed in 1 more patient (9/14, 64,28%, not statistically significant with Fisher’s exact test, *p* = 0.51). The total required dose of levothyroxine was increased in the group of hypothyroid patients during the study (1.08 ± 0.26 before treatment, 1.14 ± 0.19 in the first year, and 1.37 ± 0.49 after 10 years, expressed as μg/kg) without reaching statistical significance. In the two patients who developed hypothyroidism due to RAI treatment within the first-year of following up, the thyroxin dose (μg/kg) increased from 0.90 ± 0.10 to 1.01 ± 0.29 by the end of the follow-up period.

### 4.3. Adverse Effects

The proportion of patients experiencing adverse effects was small (3/14, 21%). Adverse effects were mild: one patient developed transient symptomatic thyrotoxicosis requiring β-blockers, and two patients experienced cervical discomfort/pain, typically within the first 2 weeks after 131-I therapy. Overt biochemical hyperthyroidism was observed in four patients. Thyroid-associated orbitopathy was not observed in any patient.

## 5. Discussion

The effectiveness of 131-I therapy for multinodular goiter is limited by irregular 131-I uptake in the gland, and the degree of goiter reduction is inversely correlated with the initial goiter size [21]. Therefore, individual responses vary, and high 131-I activities may be required for sufficient 131-I accumulation, especially in cases of low radioiodine uptake and very large goiters. Recombinant human thyroid-stimulating hormone (rhTSH) has been approved for thyroid cancer management and explored off-label for large goiters, increasing 131-I uptake and reducing the required 131-I dose. Pretreatment with rhTSH promotes a more uniform distribution of 131-I within the nodular gland, stimulating greater 131-I uptake in cold areas compared to hot areas [22]. When rhTSH pre-stimulation is applied alongside a conventional dose of 131-I, goiter size reduction appears to increase from approximately 40% to 60% within a year [23]. This improvement may result in more satisfactory outcomes of 131-I therapy for patients with very large goiters, who would otherwise undergo thyroidectomy.

In randomized, double-blind studies with a 12-month follow-up, goiter reduction was 62% in the rhTSH group (compared to 46% in the placebo group, *p* = 0.002) [24] and 53% in patients with large goiters (compared to 34% in the placebo group, *p* = 0.002) [25]. Few data on TV reduction are available from studies with follow-up periods longer than one year [15,26,27]. In addition, it is difficult to compare their results due to the varied treatment schedules, differences in initial TV volume, and small number of subjects. Romao et al. [14] reported an 80% reduction in euthyroid MNG subjects pre-treated with rhTSH after 36 months. Over an extended follow-up period (71 months), a further TV reduction was observed in the rhTSH group [26] compared to that detected within the first year of treatment (69.7 ± 3.1% vs. 59.2 ± 2.2%, respectively). Intriguingly, our results indicated a tendency towards a further decrease in thyroid volume 10 years after treatment; however, this trend did not reach statistical significance, probably owing to the small sample size.

In a study of 18 patients [27] with a longer follow-up (55.3 ± 4.1 months) and similar rhTSH dose to that in our report (0.1 mg of rhTSH), TV reduction was only positively correlated with initial TV, whereas we found a positive correlation between TV reduction in 4 months and that detected at the end of the follow-up period (Figure 2). This may represent a prospective prognostic tool for the procedure’s long-term efficacy and should be validated in larger cohorts. In the above-mentioned study, rhTSH was administered with a different stimulation protocol (two injections of 0.1 mg each on two consecutive days), and RAI therapy was weighted according to RAI under rhTSH pre-treatment and TV, up to the maximum allowed activity of 16.2 mCi (600 MBq). Differences in the treatment protocols were likely responsible for the disparate outcomes compared to our results.

We observed a significant reduction in the diameter of the larger nodules following RAI therapy after one year of observation. However, there was no further significant decline thereafter. After the investigation, however, the diameter of the nodules in some patients increased. Since there are no data in the literature, we presume that the initial positive effect should be attributed to the destructive effect of RAI on adjacent healthy thyroid tissue, which resulted in a beneficial effect on nodule size. Importantly, no new nodules developed in any patient during the follow-up, indicating a possible protective effect of RAI therapy. This potential long-term effect has not yet been investigated in the treatment of MNG with rhTSH, even though treatment with high doses of 131-I increases the likelihood of cancer induction. A modest dose of RAI, such as that administered to our patients, would likely reduce the concern of malignancy development. Once more, these findings must be validated in a larger study population.

Long-term hypothyroidism may follow transient hyperthyroidism, and our study was the first to report data after a decade of RAI treatment. During these 10 years of following up, three out of eight (37.5%) of our patients developed hypothyroidism. It was reported that 56% of patients who received a larger dose of radioiodine (30 mCi) with rhTSH augmentation developed overt hypothyroidism within one year [28]. Fast et al. [26] also reported a long-term increase in the incidence of hypothyroidism (a prevalence of hypothyroidism of 43% at 1 year and 52% at 71 months of follow-up, substantially greater than that detected in the group treated without rhTSH stimulation). The results were difficult to interpret due to the various treatment schedules, initial TV volume differences, RAI and rhTSH doses, and the small sample size. Intriguingly, pretreatment thyroid function appeared to play a crucial role, given that in the study by Romao et al. [15], patients who were euthyroid at baseline exhibited a greater persistent hypothyroidism and TV reduction than those with subclinical or overt hyperthyroidism.

In our study, the combination of rhTSH and radioiodine was well-tolerated; however, several authors have reported that patients may experience transient destructive thyrotoxicosis with excruciating transient thyroiditis, thyroid enlargement, tracheal compression, and, frequently, heart-related symptoms [29]. As required, the administration of glucocorticoids and beta-blockers may aid in alleviating these symptoms, as needed in one of our patients. Following our findings, it has been hypothesized [30] that these adverse events may be dependent on the dose of rhTSH and are inconsequential with lower doses of stimulation, thereby making low-dose rhTSH intervention more desirable in clinical practice. We acknowledge that the present study had some limitations. Firstly, this was an observational study and not a randomized controlled trial comparing the safety and efficacy of 131-I therapy after pretreatment with rhTSH to that of standard 131-I therapy, i.e., without pretreatment with rhTSH. Second, this investigation was conducted in a region with adequate iodide intake, and the effect of rhTSH pretreatment in regions with higher or lower iodide intake than that in Northern Greece remains uncertain. Third, the limited number of patients treated undermined the validity of the results. However, the homogeneous patients and treatment characteristics (thyroid disease, RAI dose, and rhTSH dose) in our study suggested that the absence of goiter recurrence after a decade of the initial RUI treatment was a promising result and thus should be confirmed in larger cohorts.

## 6. Conclusions

Although radiation regulations may vary from country to country, an important clinical implication of our study was that a substantial reduction in required 131-I activity when rhTSH was used enabled all patients to be treated as outpatients. In addition, patient restrictions were less stringent when 131-I activity was minimal. Although a cost-effectiveness analysis was not included in this study, our findings suggest that rhTSH-augmented 131-I therapy may be a cost-effective alternative to reoperation. The majority of patients reported relief from goiter-related pressure symptoms, which was the intervention’s primary objective. It is significant to note that FNA did not completely rule out thyroid malignancy and therefore, the presence of thyroid nodules remained a concern when administering RAI therapy. Importantly, given the potential long-term adverse effects of radioiodine therapy, including the risk of malignancy, our study found no new cases of thyroid cancer during a 10-year follow-up period, despite examining a small cohort of treated patients.

Reduction in goiter volume with rh TSH-assisted RAI therapy in relapsing multinodular goiters may be considered an option for specifically selected patients. This includes individuals with comorbidities where reoperation might be challenging or not feasible.

## Figures and Tables

**Figure 1 diagnostics-14-00946-f001:**
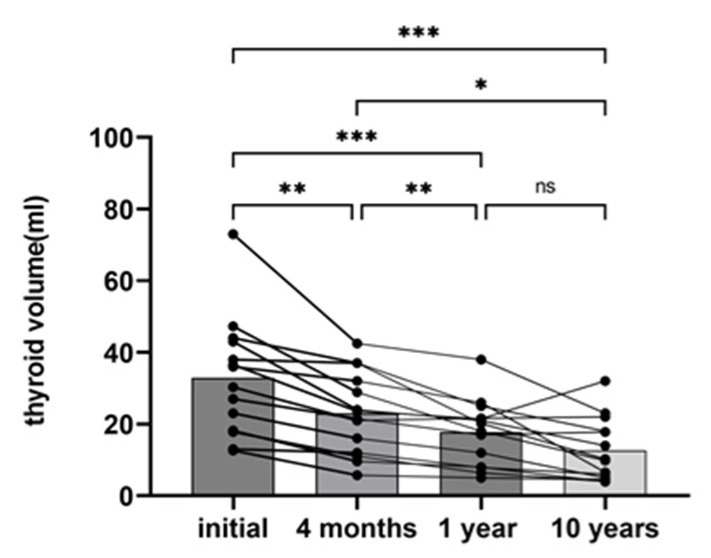
Thyroid volume during follow-up. One-way analysis of variance (ANOVA) parametric test. ***: *p* < 0.0001, **: *p* < 0.001, *: *p* < 0.05, ns: no statistical significance. Columns represent mean values in milliliters.

**Figure 2 diagnostics-14-00946-f002:**
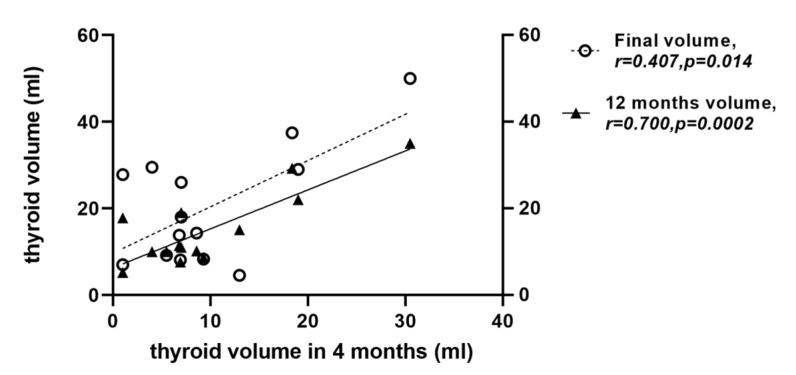
Correlations of thyroid volume. Linear regression analysis with Spearman’s non-parametric test. Circle plot: final measurement; triangle plot: measurement at 1 year.

**Figure 3 diagnostics-14-00946-f003:**
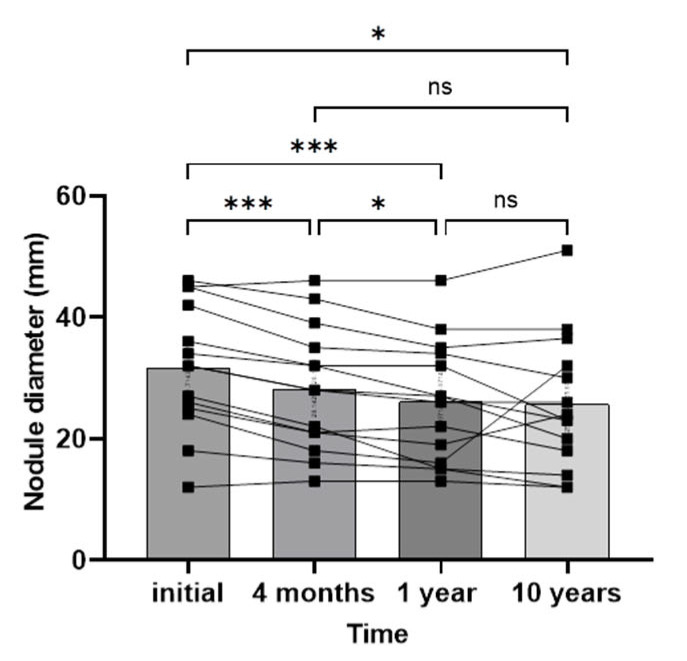
Maximum diameter of the dominant thyroid nodule during follow-up. One-way analysis of variance (ANOVA) parametric test. ***: *p* < 0.0001, *: *p* < 0.05, ns: no statistical significance. Columns represent mean values in millimeters.

**Table 1 diagnostics-14-00946-t001:** Study population characteristics.

*N* = 14	Baseline	Final	*p*
Age	59.14 ± 15.44	69.42 ± 14.98	
BMI (kg/m^2^)	27.34 ± 4.14	26.84 ± 4.69	ns
Hypothyroidism	6	9	ns
Thyroxin dose (μg/kg)	1.087 ± 0.26	1.37 ± 0.49	ns
Thyroid volume (mL)	32.16 ± 16.66	12.62 ± 8.76	<0.001
Nodule diameter (mm)	31.71 ± 10.46	25.67 ± 11.05	<0.05

Legend: *N*—number of patients, BMI—body mass index, ns—no statistical significance, *p*—*p* value with Wilcoxon non-parametric test. Data presented as mean ± standard deviation.

**Table 2 diagnostics-14-00946-t002:** Comparison of thyroid volume changes during follow-up.

	Mean Diff.	95% CI of Diff.	Adjusted *p* Value
Baseline vs. 4 months	8.37	0.76 to 15.97	0.03
Baseline vs. 1 year	14.49	7.02 to 21.94	<0.001
Baseline vs. 10 years	19.54	8.32 to 30.76	0.001
4 months vs. 1 year	6.11	2.24 to 9.98	0.002
4 months vs. 10 years	11.17	3.23 to 19.11	0.006
1 year vs. 10 years	5.05	−0.69 to 10.81	0.09

Legend: comparisons between groups’ means of thyroid volume (mL) with Tukey’s multiple comparison test. Mean Diff.—difference in means between groups; 95% CI of Diff.—95% confidence interval.

**Table 3 diagnostics-14-00946-t003:** Comparison of the size of the dominant thyroid nodule.

	Mean Diff.	95% CI of Diff.	Adjusted *p* Value
Baseline vs. 4 months	3.57	1.68 to 5.46	<0.001
Baseline vs. 1 year	5.64	2.57 to 8.71	<0.001
Baseline vs. 10 years	6.03	0.50 to 11.57	0.03
4 months vs. 1 year	2.07	0.24 to 3.90	0.03
4 months vs. 10 years	2.46	−2.61 to 7.54	0.51
1 year vs. 10 years	0.39	−4.44 to 5.23	>0.99

Legend: comparisons between groups’ means of the diameter of the dominant nodule (mm) with Tukey’s multiple comparison test. Mean Diff.—difference in means between groups; 95% CI of Diff.—95% confidence interval.

## Data Availability

The datasets are available from the corresponding author upon reasonable request.

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
