# Peer review of "Long-Term Effects of 0.1 mg Recombinant-Human-Thyrotropin-Stimulated Fixed-Dose Radioiodine Therapy in Patients with Recurrent Multinodular Goiter after Surgery"

_diagnostics, 2024, doi:10.3390/diagnostics14090946_

Round 1
Reviewer 1 Report
Comments and Suggestions for Authors
The paper is a follow-up manuscript of a previous paper published by the authors in 2012. While the idea of I131 administration in relapsing euthyroid nodular goiters may be attractive from some points of view, there are some major issues to be addressed:
1. Introduction: LT4 suppressive therapy for euthyroid nodular goiter is no longer endorsed by international guidelines (2023 ETA guidelines for thyroid nodule management), therefore no longer an option in these patients. Also, current guideline indications in these patients should be mentioned, along with indications for subtotal thyroidectomy. Surgical reintervention usually occurs due to malignancy concerns or compressive symptoms. Line 35: " with no cosmetic symptoms"? Maybe the authors intended to say "compressive symptoms"?
Results and Discussion: It should be mentioned that FNAB does not 100% exclude thyroid malignancy, therefore this remains among the concerns of administering RAI when thyroid nodules are present.
Line 150: the values of TV at baseline and at 10 years, respectively, are inversed. Line 151: no statistical differences was "further" reached.
Line 221: indicated "tendency towards " a further decrease in thyroid volume 10 years after treatment
Line 230-231: please clarify, hard to follow
How would the authors interpret the rise in nodule diameter after RAI? In how many cases was this encountered? Was FNAB reperformed afterwards?
It should be noted that, overall, thyroid volume reduced more than nodule diameter and the greatest effect was registered 1 year after RAI in both. Thus, 10 year follow up is rather important considering long-term adverse effects, especially malignancy risk. This should be discussed.
Also, other studies reported rhTSH-RAI administration in much larger goiters (volume > 70-100 ml). This should be mentioned.
Finally, reduction in goiter volume with RAI in relapsing multinodular goiters may become an option in specifically selected patients- maybe the authors may specifically point out such situations?
English language fine, no special concerns.
Author Response
Introduction:
- LT4 suppressive therapy for euthyroid nodular goiter is no longer endorsed by international guidelines (2023 ETA guidelines for thyroid nodule management), therefore no longer an option in these patients. Also, current guideline indications in these patients should be mentioned, along with indications for subtotal thyroidectomy. Surgical reintervention usually occurs due to malignancy concerns or compressive symptoms.
Response: Changes have been made to clarify LT-4 therapy and surgery treatment options.
- Line 35: " with no cosmetic symptoms"? Maybe the authors intended to say "compressive symptoms"?
Response: Corrected
Results and Discussion:
- It should be mentioned that FNAB does not 100% exclude thyroid malignancy, therefore this remains among the concerns of administering RAI when thyroid nodules are present.
Response: This is mentioned in the discussion section.
- Line 150: the values of TV at baseline and at 10 years, respectively, are inversed.
Response: Corrected
- Line 151: no statistical differences was "further" reached.
Response: Corrected
- Line 221: indicated "tendency towards " a further decrease in thyroid volume 10 years after treatment
Response: Corrected
- Line 230-231: please clarify, hard to follow
Response: Rephrased
- How would the authors interpret the rise in nodule diameter after RAI? In how many cases was this encountered? Was FNAB reperformed afterwards?
Response: That was observed in two patients, and we added this information to the results. Fna was performed to rule out malignancy.
- It should be noted that, overall, thyroid volume reduced more than nodule diameter and the greatest effect was registered 1 year after RAI in both. Thus, 10 year follow up is rather important considering long-term adverse effects, especially malignancy risk. This should be discussed.
Response: We highlighted this finding in the conclusion section.
- Also, other studies reported rhTSH-RAI administration in much larger goiters (volume > 70-100 ml). This should be mentioned.
Response: A more extensive discussion was made about the effectiveness of rhTSH-assisted RAI therapy.
- Finally, reduction in goiter volume with RAI in relapsing multinodular goiters may become an option in specifically selected patients- maybe the authors may specifically point out such situations?
Response: We agree and point it out in the conclusion.
Reviewer 2 Report
Comments and Suggestions for Authors
General comments
The study by Angelopoulos et al entitled "Long-term Effects of 0.1 mg Recombinant Human Thyrotropin-Stimulated Fix Dose of Radioiodine Therapy in Patients with Recurrent Multinodular Goiter after Surgery" is a clinical interventional retrospective study which aimed to determine the durability of this beneficial effect in a solid cohort of patients with recurrent euthyroid nontoxic multinodular goiter after partial thyroidectomy, who were initially treated with a fixed small dose of RAI (11mCi) under 0.1 mg of rhTSH stimulation 10 years after therapy.
Major comments
The study is interesting and its findings have useful clinical implications.
Minor comments
1) Lines 81-85 of the Introduction session should be done as a separate session entitled AIM
2) Στα Results, lines 148-150, the data for the thyroid volume, before and after the treatment, are presented in reverse and the same for the nodules (lines 174-176). The same mistake happens in Table 1.
3) In Results-Thyroid function session the details of the values of the Thyroid Function Tests, of the non-hypothyroid patients, at baseline should be written (mean +SD). The possible presence of near normal low TSH values (indicative of hypotoxic multinodular goiter), indicates that some nodules in the remnant suggests are hot. This could explain the wide variation in nodule size regression after RAI, because hot and cold nodules are also affected by rTSH (Bountouris et al Frontiers in Endocrinology 2023).
4) In Conclusion session the 2nd paragraph, lines 277-286, should be moved at the end of the Discussion session.
Author Response
Minor comments
- Lines 81-85 of the Introduction session should be done as a separate session entitled AIM
Response: Modified in a separate section
2) Results, lines 148-150, the data for the thyroid volume, before and after the treatment, are presented in reverse and the same for the nodules (lines 174-176). The same mistake happens in Table 1.
Response: Corrected
- In Results-Thyroid function session the details of the values of the Thyroid Function Tests, of the non-hypothyroid patients, at baseline should be written (mean +SD). The possible presence of near normal low TSH values (indicative of hypotoxic multinodular goiter), indicates that some nodules in the remnant suggests are hot. This could explain the wide variation in nodule size regression after RAI, because hot and cold nodules are also affected by rTSH (Bountouris et al Frontiers in Endocrinology 2023).
Response: Data for baseline thyroid function tests are added in the results section.
4) In Conclusion session the 2nd paragraph, lines 277-286, should be moved at the end of the Discussion session.
Response: We agree and we moved the paragraph at the end of the Discussion session
Round 2
Reviewer 1 Report
Comments and Suggestions for Authors
The authors significantly improved the manuscript. The Methodology, Discussion and Conclusion sections are appropriate, highlighting specific advantages of the proposed manner of treatment.